# Wealth-based inequality in the exclusive use of hygienic materials during menstruation among young women in urban India

**Aditya Singh**[1]*, **Mahashweta Chakrabarty**[1], **Shivani Singh**[2], **Diwakar Mohan**[3], **Rakesh Chandra**[4], **Sourav Chowdhury**[5]

**1** Banaras Hindu University, Varanasi, Uttar Pradesh, India, **2** India Health Action Trust, Lucknow, Uttar Pradesh, India, **3** Johns Hopkins Bloomberg School of Public Health, Baltimore, Maryland, United States of America, **4** Tata Institute of Social Science, Mumbai, Maharashtra, India, **5** Raiganj University, Raiganj, West Bengal, India

* adityasingh@bhu.ac.in

**Data Availability Statement:** All unit level datasets used in this study are available on request and free of cost from the Demographic and Health Surveys (DHS) Program website https://dhsprogram.com/.

## Abstract

### Background

The exclusive use of hygienic materials during menstruation (sanitary napkins, locally made napkins, tampons, and menstrual cups) among urban women in India has been increasing over time. However, little is known about the wealth-based disparity in the exclusive use of hygienic materials during menstruation among these women. This study, therefore, measures wealth-based inequality in the exclusive use of hygienic materials during menstruation among urban women in India. Furthermore, the measured inequality is decomposed to unravel its contributing factors.

### Data and methods

Using data from the National Family Health Survey-5 (2019–21), we calculated the Erreygers normalized concentration index (CI) for India and each of its states to measure wealth-based inequality in the exclusive use of hygienic materials during menstruation among women in urban India. Further, we decomposed the Erreygers CI to estimate the relative contribution of covariates to wealth-based inequality in the exclusive use of hygienic materials during menstruation. The analysis included 54,561 urban women aged 15–24 from 28 states and eight union territories of India.

### Results

The Erreygers CI value of 0.302 indicated a pro-rich inequality in the exclusive use of hygienic materials among urban women in India. While all the states and UTs showed pro-rich inequality, the CI varied considerably across the country. Among the bigger states, the inequality was highest in Madhya Pradesh (CI: 0.45), Assam (CI: 0.44), Bihar (CI: 0.41), and West Bengal (CI: 0.37) and the lowest in the south Indian states of Tamil Nadu (CI: 0.10), Andhra Pradesh (CI: 0.15), Telangana (CI: 0.15), and Kerala (CI: 0.20). Erreygers decomposition revealed that wealth-based inequality in women's education and mass media

**Funding:** The author(s) received no specific funding for this work.

**Competing interests:** The authors have declared that no competing interests exist.

exposure contributed almost 80% of the wealth-based inequality in the exclusive use of hygienic materials during menstruation among urban women in India.

## Conclusion

Substantial pro-rich inequality in the exclusive use of hygienic materials suggests that the policies and program initiatives should prioritize reaching out to poor women to increase the overall rate of exclusive use of hygienic materials during menstruation in urban India.

## Introduction

Most of the developing countries, including India, are witnessing rapid urbanization recently. India's urban population has increased from 17.0% in 1951 to 34.9% in 2020 (285 million), with an annual growth of 2.3% between 2015 and 2020 [1]. Studies suggest that rapid urbanization in developing world is associated with widening income inequality among urban population [2–5]. It is also observed that living conditions and health status of the urban poor could be remarkably different from others of the urban poor could be remarkably different from others in the same city [6]. Urban poor have to live in areas lacking basic amenities such as sanitation, running water, and drainage, and suffer unhealthy and unclean living conditions [7]. This unequal access to resources, coupled with inequality in the provision of health services, often manifests itself in pervasive health inequalities in urban areas [2, 8].

Due to the patriarchal structure of Indian society, it is unsurprising that urban poor women have to suffer more than men [9]. Culturally, women in most Indian households are expected to balance household income and needs. In the process, the needs and demands of other household members are prioritized, and women have to compromise their health and hygiene needs (including menstrual needs), which negatively impacts their reproductive health [6, 10]. Gynaecological diseases are the leading cause of disease burden among women of reproductive age in India, contributing about 6% of the total disability-adjusted life years (DALYs) [11]. To reduce this burden, menstrual hygiene management among women is important, as menstruation is the first biological process through which women enter into reproductive cohort [12]. However, menstrual health has been largely neglected in reproductive health research and deserves special attention due to its significant contribution to the disease burden among women.

In India women use materials such as clothes, sanitary napkins, tampons, menstrual cups, etc., to prevent menstrual bloodstains from being visible. In general, these products are grouped into two categories: hygienic materials (sanitary pads, tampons, menstrual cups) and unhygienic materials (clothes, old rags, socks, newspaper, dried leaves, etc.) [13]. Ensuring universal access to hygienic materials promotes better reproductive health and improved outcomes across diverse aspects of human development [14]. In the past two decades, various sanitary movements, awareness campaigns, and initiatives to distribute subsidized sanitary napkins in urban areas have led to an increase in the use of hygienic materials among urban women in India [15]. However, little is known about the extent to which this increase has been equitable or inequitable across different wealth groups in different states of India.

A number of studies in the recent past have attempted to understand the knowledge, attitude and practices regarding hygienic materials of protection during menstruation among adolescent girls in urban India [16–24]. It is reported that there are considerable differences in urban women's choice of menstrual absorbents based on socioeconomic and biodemographic

variables in India [21, 25–30]. However, most of these studies are limited in their geographical scope. None of them has attempted to measure wealth-based inequality in the exclusive use of hygienic materials during menstrual period among urban women at national and state level. Understanding wealth-based inequalities in the exclusive use of hygienic materials during menstruation among young urban women should be an integral part of any effort to promote equity across the states of India. This study, therefore, aims to measure the wealth-based inequality in the exclusive use of hygienic materials among women in India and each of its states and decomposes this inequality in its contributing factors.

## Data and methods

### Data source

We used data from the latest round of the National Family Health Survey (NFHS-5) conducted between 2019 and 2021. The NFHS is a large, nationally representative survey that provides data on India's population, health, and nutrition status [31]. The details of the sampling procedure used in the survey and response rates are available in the national report [31]. NFHS-5 interviewed 724,115 women of reproductive age (15–49 years) from 636,699 households. 544,580 rural women were excluded due to our focus on urban women in India. Of these 179,535 remaining women, 124,974 urban women aged 25–49 were excluded because of missing menstrual hygiene data points. The remaining 54,561 urban women aged 15–24 from 28 states, eight union territories and 707 districts were included in the analysis.

### Ethical statement

This study only used anonymous public-use dataset provided by the Demographic and Health Surveys (DHS) program for which ethical approval is not required. The survey data used in this study can be obtained by making a formal request on the official website of the DHS program (https://dhsprogram.com/data/available-datasets.cfm).

### Statistical analysis

Initially, we calculated the sample distribution of urban women aged 15–24 by their background characteristics. Then, we used the Chi-square test to examine the significant association between the exclusive use of hygienic materials and socioeconomic and demographic indicators. Next, we used the Erreygers Concentration Index (CI), a method proposed by Erreygers, to measure wealth-based inequalities in the use of hygienic materials in each state of India [32]. Lastly, Erreygers CI is decomposed to explore the socioeconomic and demographic factors contributing to inequalities in using hygienic materials during menstruation. We conducted all the analyses in Stata 16 statistical software [33]. We used the "svyset" command in Stata statistical software to adjust for the complex sample design of the NFHS.

**Erreygers concentration index.** The concentration index measures and compares the degree of socioeconomic-related inequality in health sector variables. The concentration index is twice the area between the concentration curve and the line of equality [34].

The standard concentration index of health C(h)equal to

$$C(h) = \frac{2}{\mu} cov(h_i, \, r_i) \tag{1}$$

It can also be written as [34],

$$C(h) = \frac{2}{n\mu} \sum\nolimits_{i=1}^{n} h_i r_i - 1 \tag{2}$$

Where, $h_i$ is the health sector variable, $r_i$ is the fractional rank of individual $i$ in the living standard distribution, μ is the health sector variable's mean, and n is the sample size. The concentration index ranges between -1 and +1. The negative value of CI indicates a disproportionate distribution of the health variable among the poor; in this case, the concentration curve lies above the line of equality. Conversely, the index takes a positive value when the concentration curve lies above the equality line, indicating the health variable concentration among the rich. The concentration index is zero when there is no inequality.

When the health sector variable is binary, the standard concentration index C(h) misestimates the extent of inequality because the bounds of the CI shrink with the increase in the mean of the health sector variable [32, 35]. According to Wagstaff, in the case of a binary health variable, the CI should be normalized by dividing it either by the reciprocal of the mean of the variable or the bound of the CI [35]. Also, Wagstaff's CI does not satisfy all the four conditions (i.e., mirror, transfer, level independence, cardinal invariance) of rank-dependent indices [36]. To make a good rank-dependent socioeconomic indicator of health that satisfies all the four properties of a rank-dependent index mentioned above, Erreygers has defined a corrected CI for binary variables as [32]:

$$E(h) = \frac{8\mu}{n^2(b_h - a_h)} \sum_{i=1}^{n} z_i h_i$$

Where, where $h_i$ is a binary variable that is equal to 1 when the woman uses hygienic materials exclusively and 0 otherwise; $z_i = \frac{(n+1)}{2} - \lambda_i$ where n is the sample size, and $\lambda_i$ denotes the socioeconomic rank of the individual ranging from the richest ($\lambda_i = 1$) to the poorest ($\lambda_i = n$), $b_h$ is the maximum value of health variable and $a_h$) is the lowest value of health variable.

**Decomposition analysis.** Wagstaff proposed a decomposition of CI for a linearly additive regression model of the health variable of interest. This linear decomposition analysis cannot be used on binary health variables. [37, 38]. To solve this problem, Doorslaer and others suggested the 'marginal effects' $(\frac{d_y}{d_x})$ approximately restored the mechanism of the decomposition framework [38–40]. The main drawback of Doorslaer's decomposition was that it was based on the original formula of standard CI, which had flaws discussed earlier [35, 38]. Fortunately, these technical loopholes of decomposition analysis were addressed by Erreygers [32]. Erreygers has given a corrected decomposition technique for corrected concentration index as:

$$E(h) = 4\left[\sum_{j=1}^{q} \theta_j^* V\left(x_j\right) + V(e^*)\right]$$

Where $\theta_j^*$ are the coefficients of independent variables, $V(x_j)$ is the generalised concentration index of $x_j$ and $V(e^*)$ is the generalised concentration index of error terms ($e^*$). For the steps involved in the Erreygers decomposition, please refer to Erreygers (2009) [32]. Erreygers normalised CI has been chosen to be applied in this study because it fulfils four basic properties of CI, i.e., the principles of mirror, transfer, cardinal invariance, and level independence [36].

**Outcome variable.** NFHS-5 asks a multiple-response question to eligible female respondents about materials used during their menstrual period to prevent blood stains from becoming evident. Response options included seven categories, i) locally made napkins, ii) sanitary napkins, iii) tampons, iv) menstrual cups, v) cloth, vi) nothing, and vii) others. The NFHS-5 categorises these materials into two: hygienic and unhygienic. The first four of these are labelled as hygienic materials, and the remaining as unhygienic [31].

The outcome variable of this study is "exclusive use of hygienic materials". It is a binary variable. A woman is considered "an exclusive user of hygienic materials" if she uses hygienic materials only. This category was coded as '1'. Any woman who either uses unhygienic materials or a combination of hygienic and unhygienic materials is considered "not an exclusive user of hygienic materials". This category was coded as '0'. This variable has been defined in this way in many previous studies in India [25, 41–43].

**Predictor variables.** It is well established that disposable income or household wealth is an important predictor of using hygienic materials during menstruation [25, 26, 28]. Since the NFHS did not collect household income or expenditure data, this study used the household wealth index given in the dataset as a proxy for a household's economic status. The household wealth index is computed based on economic proxies, such as housing quality, household amenities, consumer durables, and land holding size [44]. It is calculated using principal component analysis (PCA) and divided into five quintiles—poorest, poorer, middle, richer, and richest.

We considered a range of socioeconomic and biodemographic predictors, including age at menarche, child marriage, respondent's education level, religion, social groups, type of home, and mass media exposure. The existing literature guides the choice of variables for menstrual hygiene management [25–28, 41, 42]. Detailed information and categorization of variables used in the analysis are given in Table 1.

## Results

### Sample characteristics

Table 2 shows the background characteristics of 54561 sampled women aged 15–24 from urban India. Only a minuscule proportion of urban women did not complete their secondary education. Approximately one-third of the women in the sample were SC/ST, and little over two-thirds were Hindu. A significant proportion of urban women had at least one type of mass media exposure. Nearly half women were from poor households.

### Exclusive use of hygienic materials during menstruation by background characteristics

Using NFHS-5 data, we examined differences in the exclusive use of hygiene materials among sampled urban women stratified by socioeconomic status. Table 2 shows that a significant proportion of Hindu women compared to non-Hindu women reported using hygienic materials during menstruation. Exclusive use of hygiene materials was considerably higher among those exposed to mass media than those not exposed to any form of mass media. Exclusive use was higher among those who married after 18 years of age.

### Spatial variation in the exclusive use of hygienic materials

We found substantial regional and state-level variation in the exclusive use of hygienic materials during menstruation in urban India (see Figs 1 and 2). Among the regions of India, the exclusive use was highest in the southern and northern regions and lowest in the central region (50.9%). At the state-level, the exclusive use was highest in Mizoram (92.6%) and lowest in Manipur (43.6%). Exclusive use of hygienic materials during menstruation in urban areas of Bihar, Chhattisgarh, and Uttar Pradesh was less than 50%. This proportion was over 90% in the state of Tamil Nadu. As for the UTs, the exclusive use was highest in Andaman and Nicobar (94.2%) and lowest in Ladakh (52.2%).

**Table 1. Operational definition of variables.**

| Variables | Definition |
|---|---|
| Age at menarche (in years) | Age at menarche indicates the age of onset of first menstrual period of a women. 15–24 years urban women who received their first period at 13 years and above is coded as "1" and if the women received her first period before 13 years, then coded as "0". |
| Education | Education is categorised as a binary variable. 15–24 years urban women who have completed their education till secondary level and above is coded as "1" and a "0" if did not reach secondary completion or below. |
| Religion | Religion is categorised into two categories. Household belonging to Hindu religion is coded as "1", and "0" if belonging from non-Hindu religion. |
| SC/ST | SC/ST is a categorical variable having two categories. Women belonging to Scheduled Caste or Scheduled Tribe social groups is coded as "1" and "0" if belonging from OBC or other categories. |
| Mass media exposure | Three questions were asked to women in NFHS-5 survey. They are i) how often they read newspaper/magazines, ii) how often they watch television, and iii) how often they listen to radio. The responses are 'almost every day', at least once a week, less than once a week and not at all.<br>Women who have exposure of at least any one type of mass media is considered as having mass media exposure and coded as "1", and "0" otherwise. |
| Child marriage | 15–24 years urban women who is married before reaching the legal age of marriage, i.e., 18 years is coded as "1" and "0" otherwise. |
| Type of home | On the basis of the woman's relationship with the household head, the type of home in which she resided was categorised as 'marital home' (wife, daughter-in-law, or sister-in-law of the household head) coded as "1" and coded as "0" otherwise (daughter, granddaughter, or niece of the household head, nonrelatives such as domestic servants working in the household, orphans, deserted young women). |
| Northern region | Household in northern region of India is coded as "1" (includes Jammu & Kashmir, Ladakh, Himachal Pradesh, Punjab, Rajasthan, Haryana, Uttarakhand, Chandigarh and Delhi) and "0" otherwise. |
| Central region | Household in central region of India is coded as "1" (includes the states of Uttar Pradesh, Madhya Pradesh and Chhattisgarh) and "0" otherwise. |
| Eastern region | Household in eastern region of India is coded as "1" (includes the states of Bihar, Jharkhand, West Bengal and Odisha) and "0" otherwise. |
| Western region | Household in western region of India is coded as "1" (includes the states of Gujarat, Maharashtra, Goa and UTs of Dadra & Nagar Haveli and Daman & Diu) and "0" otherwise. |
| Southern region | Household in southern region of India is coded as "1" (includes the states of Kerala, Karnataka, Andhra Pradesh, Tamil Nadu and the UTs of Andaman & Nicobar Islands, Pondicherry and Lakshadweep) and "0" otherwise. |
| North-eastern region | Household in north-eastern region of India is coded as "1" (includes the states of Sikkim, Assam, Meghalaya, Manipur, Mizoram, Nagaland, Tripura, and Arunachal Pradesh) and "0" otherwise. |
| Household wealth | In the NFHS-5, scores are assigned to households based upon the number and types of consumer items they possess, such as a television, bicycle, or car, as well as housing attributes such as the source of drinking water, toilet facilities, and flooring materials. Principal component analysis was used to get these scores [31]. |

## Wealth disparities in the use of hygienic materials

Fig 3 shows that the exclusive use of hygienic materials increased with an increase in household wealth. While approximately 86% of women in the wealthiest quintile reported exclusive use of hygienic materials during menstruation, only 48% of women in the poorest quintile did so. The richest-poorest rate-ratio was 1.80.

A detailed picture of this rich-poor gap in the exclusive use of hygienic materials is more clearly visible in the concentration curve (see Fig 4). The CC shows the cumulative proportion of women reporting exclusive use of hygienic materials during menstruation against the

**Table 2. Descriptive statistics of sampled urban women aged 15–24 years, and percentage of women using hygienic materials exclusively by background characteristics, NFHS-5, 2019–21.**

| Background Characteristics | N (54,561) | Sample distribution %[a] | Weighted %[b] of Women Exclusively Using Hygienic materials N = 54561 | 95% Confidence Interval | |
|---|---|---|---|---|---|
| | | | | Lower | Upper |
| **Respondent's current age (in years)** | | | **(16.70)** *** | | |
| 15–19 | 26,509 | 48.59 | 68.93 | 67.75 | 70.09 |
| 20–24 | 28,052 | 51.41 | 67.30 | 66.21 | 68.37 |
| **Age at menarche (in years)** | | | **(25.56)** *** | | |
| Less than 13 | 11,240 | 20.60 | 69.17 | 67.57 | 70.72 |
| 13 and above | 43,321 | 79.40 | 68.06 | 66.99 | 69.11 |
| **Education** | | | **(713.43)** *** | | 0 |
| Below secondary level | 4,032 | 7.39 | 36.85 | 34.30 | 39.47 |
| Secondary level and above | 50,529 | 92.61 | 70.64 | 69.70 | 71.56 |
| **Religion** | | | **(74.79)** *** | | |
| Non-Hindu | 16,193 | 29.68 | 61.49 | 59.50 | 63.44 |
| Hindu | 38,368 | 70.32 | 70.32 | 69.29 | 71.32 |
| **Social groups** | | | **(20.74)** *** | | |
| Non-SC/ST | 38,275 | 70.15 | 69.08 | 67.99 | 70.16 |
| SC/ST | 16,286 | 29.85 | 65.14 | 63.54 | 66.72 |
| **Household wealth** | | | **(742.37)** *** | | |
| Non-poor | 29,427 | 53.93 | 77.50 | 76.52 | 78.45 |
| Poor | 25,134 | 46.07 | 56.09 | 54.64 | 57.53 |
| **Exposure to mass media** | | | **(371.33)** *** | | |
| No mass media exposure | 4,499 | 8.25 | 44.79 | 42.06 | 47.55 |
| At least one mass media exposure | 50,062 | 91.75 | 70.07 | 69.10 | 71.01 |
| **Age at marriage** | | | **(203.77)** *** | | |
| Before 18 years | 3,807 | 6.98 | 52.36 | 49.80 | 54.9 |
| After 18 years | 50,754 | 93.02 | 69.44 | 68.46 | 70.4 |
| **Type of home** | | | **(150.01)** *** | | |
| Marital home | 12,040 | 22.07 | 60.98 | 59.45 | 62.49 |
| Other home | 42,521 | 77.93 | 70.25 | 69.21 | 71.28 |
| **Region of residence** | | | **(2976.82)** *** | | |
| Northern | 12,893 | 23.63 | 77.97 | 76.67 | 79.23 |
| Central | 11,548 | 21.17 | 50.90 | 49.00 | 52.8 |
| Eastern | 6,612 | 12.12 | 62.71 | 59.93 | 65.42 |
| Western | 6,990 | 12.81 | 74.06 | 71.63 | 76.36 |
| Southern | 10,724 | 19.66 | 77.90 | 76.28 | 79.44 |
| North-eastern | 5,794 | 10.62 | 56.79 | 53.17 | 60.34 |

Note:

[a] = column percentage,

[b] = row percentage,

Figures in parentheses are the Chi-square statistics; Chi-square test applied for each variable, Level of significance:

***p < 0.001

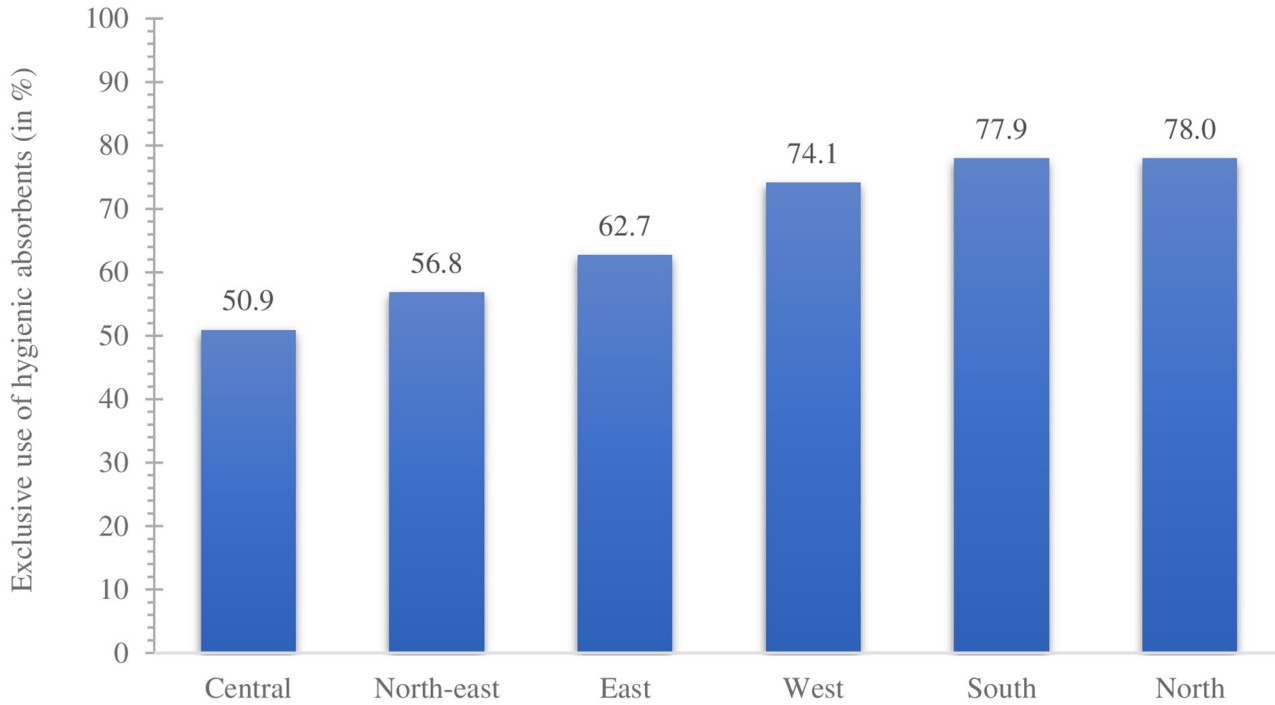

**Fig 1. Regional variation in the exclusive use of hygienic materials (%) among urban women aged 15–24 years in India, NFHS-5, 2019–21.**

cumulative percentage of the population ranked by wealth score. The CC for the exclusive use of hygienic materials lies below the line of equality, suggesting prevailing pro-rich inequality in the exclusive use. We calculated Erreygers normalized CI To summarize the wealth inequality in the exclusive use of hygienic materials in urban India. The value of CI was 0.302, suggesting that the exclusive use of hygienic materials was concentrated among women from wealthy households.

We computed Erreygers normalized CI individually for each state and UT to examine how wealth-based inequality in the exclusive use of hygienic materials varies across the states and UTs of India. A cursory examination of the Erreygers normalized CIs revealed pro-rich inequality in the exclusive use of hygienic materials in most states, but the degree of inequality varied considerably across them (see Fig 5). The Erreygers normalized CI ranged from as low as 0.08 (p<0.001) in Arunachal Pradesh to as high as 0.45 (p<0.001) in Madhya Pradesh. Apart from Madhya Pradesh, the three other states/UTs with a high inequality (Errerygers CI>0.40) in the exclusive use of hygienic materials were Assam (Errerygers CI: 0.44, p<0.001), Jammu & Kashmir (Errerygers CI: 0.43, p<0.001), and Bihar (Errerygers CI: 0.41, p<0.001). The lowest inequality (Errerygers CI <0.15, p<0.001) was noted in the southern Indian states of Tamil Nadu, Andhra Pradesh, and Telangana (for state-wise CI values see S1 Table).

We found a strong, inverse, and statistically significant relationship between the percentage of women reporting exclusive use of hygienic materials during menstruation and wealth-based inequality at the state level (r = - 0.73) (see Fig 6). States with a moderate proportion of women (50–70%) reporting exclusive use of hygienic materials had greater wealth-based inequality in the exclusive use of hygienic materials compared to those where the proportion of women reporting exclusive use of hygienic materials was high (80–100%).

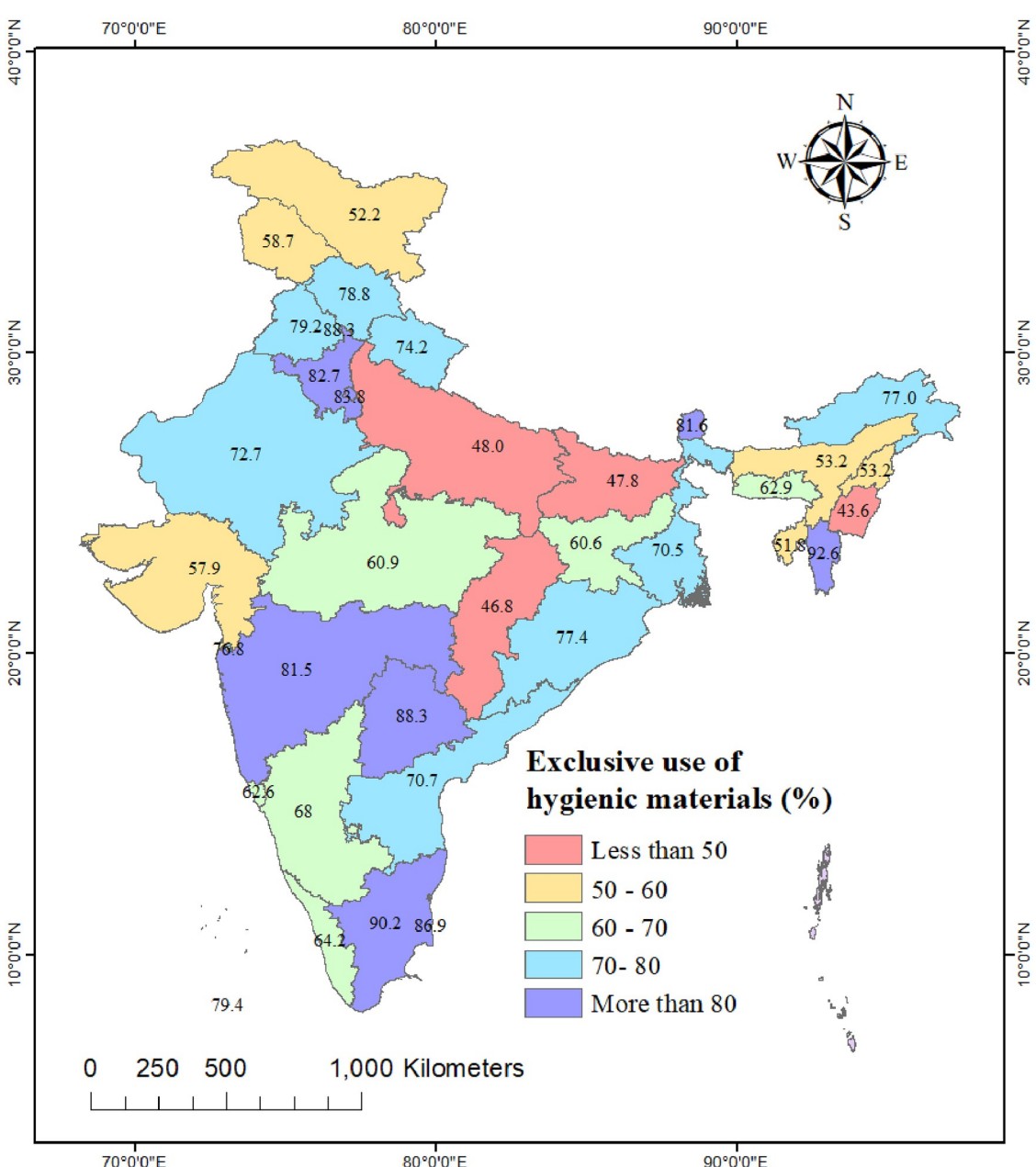

**Fig 2. Proportion of urban women aged 15–24 years reporting exclusive use of hygienic materials during menstruation in India, NFHS-5, 2019–21.**

## Decomposition of wealth-based inequality in the exclusive use of hygienic materials

Table 3 presents the results obtained from the decomposition of CI. The regression coefficients used in the decomposition of CI are reported in the second column, while the third and fourth columns report the CI and the contribution of the CI. Finally, the relative contribution of each explanatory variable in explaining inequality in the exclusive use of hygienic materials is presented in the fifth column.

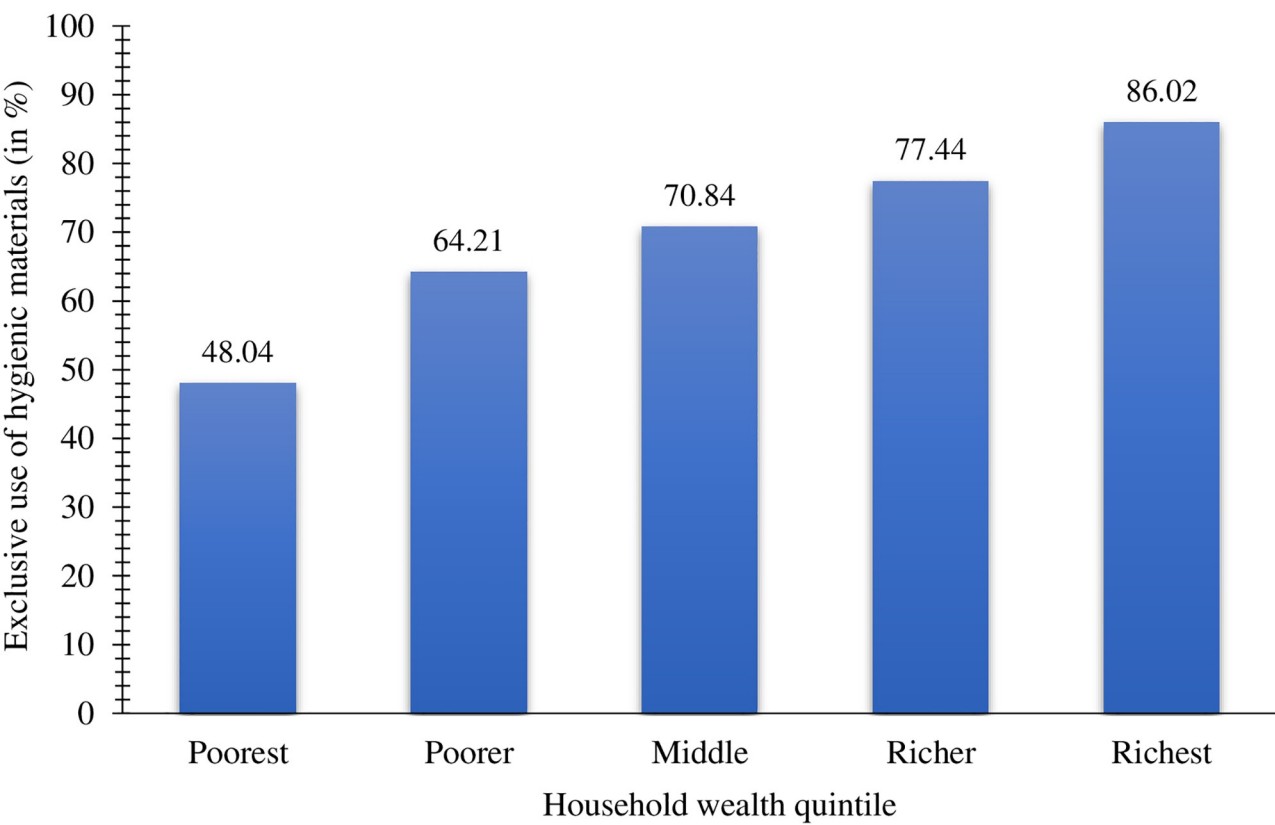

**Fig 3. Prevalence of exclusive use of hygienic materials by household wealth quintiles among urban women aged 15–24 years in India, NFHS-5, 2019–21.**

The third column of Table 3 suggests that having education above the secondary level, mass media exposure, and being Hindu are associated with increased exclusive use of hygienic materials. On the other hand, getting married before 18 years, residing in a marital home, and residing in the central region of India were associated with a decrease in the exclusive use of hygienic materials.

Four variables, i.e., education, mass media exposure, region of residence, and social groups, together explained 92% of the wealth-based inequalities in the exclusive use of hygienic materials. Education and exposure to mass media were the most critical factors, contributing about 48% and 32% of the inequality of exclusive use of hygienic materials, respectively. Wealth-based inequality in education and mass media exposure explained more than 80% of inequality in the outcome. Region of residence and social groups contributed another 10% of the inequality.

## Discussion

We assessed the level of wealth-based inequality in the exclusive use of hygienic materials among young women in urban areas and how it differed from state to state in India. We also explored the causes of this inequality using a decomposition method. We found that the exclusive use of hygienic materials in urban India was disproportionately concentrated among women from relatively wealthy households. However, the level of wealth-based inequality in the exclusive use of hygienic materials varied significantly across Indian states. The level of

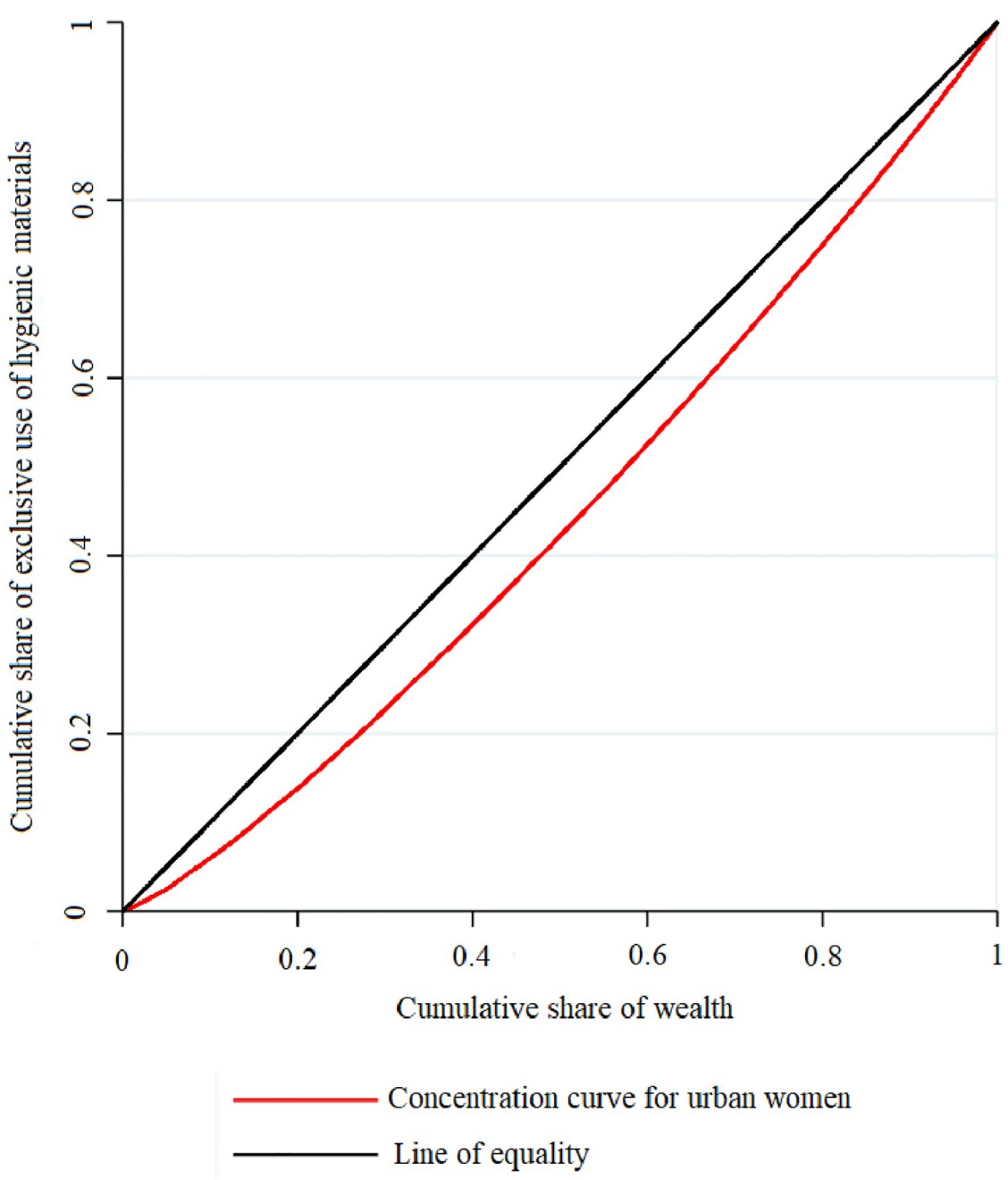

**Fig 4. Concentration Curve of exclusive use of hygienic materials among urban women aged 15–24 years in India, NFHS-5, 2019–21.**

inequality was highest in the central Indian state of Madhya Pradesh and the lowest in the southern Indian state of Tamil Nadu and the north-eastern state of Mizoram. In most cases, states with higher levels of inequality in the exclusive use of hygienic materials also had poorer coverage of the exclusive use of hygienic materials. The decomposition of CI demonstrated that most wealth-based inequality in the exclusive use of hygienic materials was attributable to

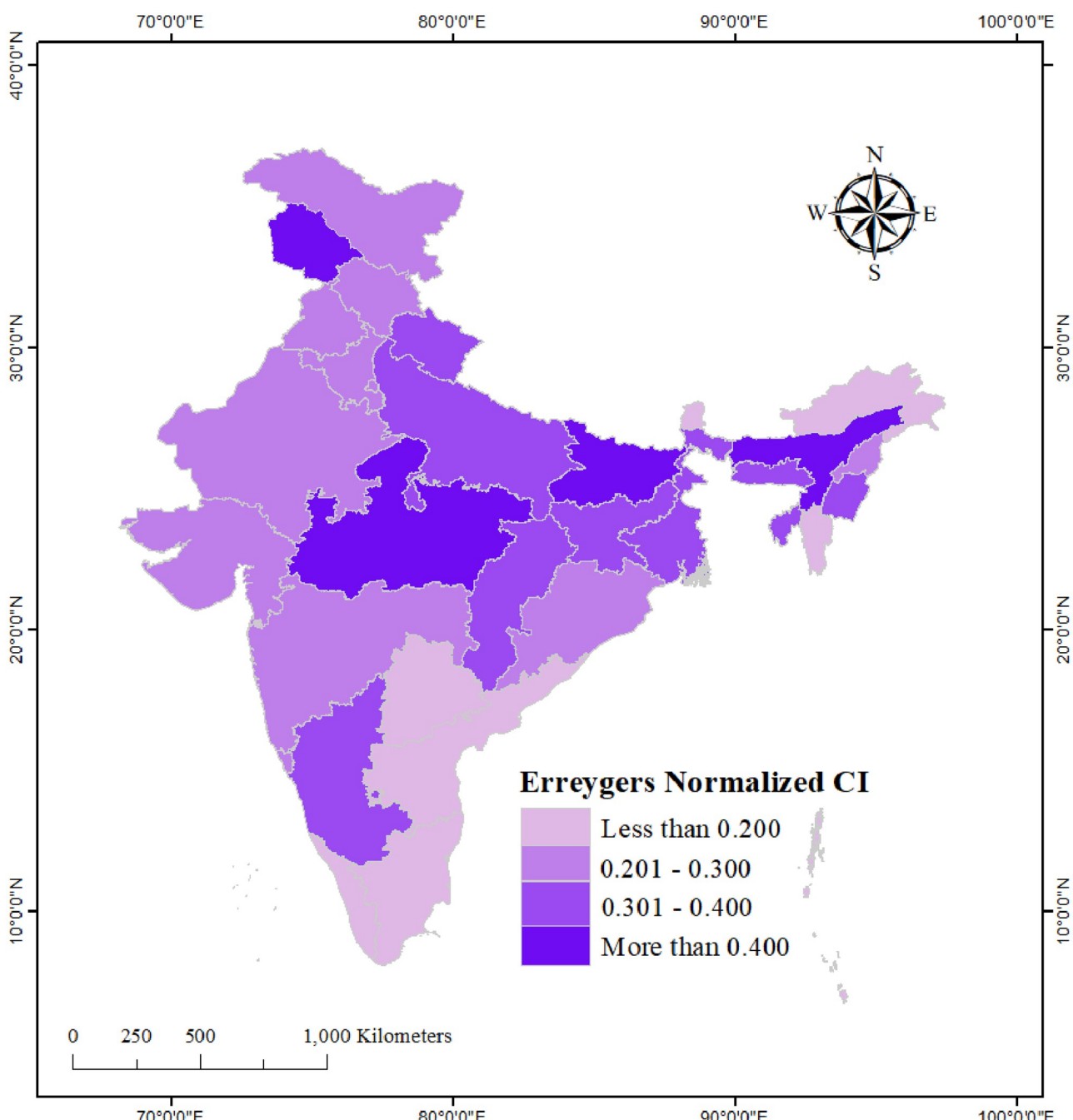

**Fig 5. State-wise wealth-based inequality in the exclusive use of hygienic materials among urban women aged 15–24 years in India, NFHS-5, 2019–21.**

wealth inequalities in women's education, mass media exposure, social groups, and region of residence.

The prevalence of exclusive use of hygienic materials progressively increased with the increase in women's education. Many previous studies in India (24,27) and other countries (42,43) have found that women's education is one of the most critical determinants of hygienic

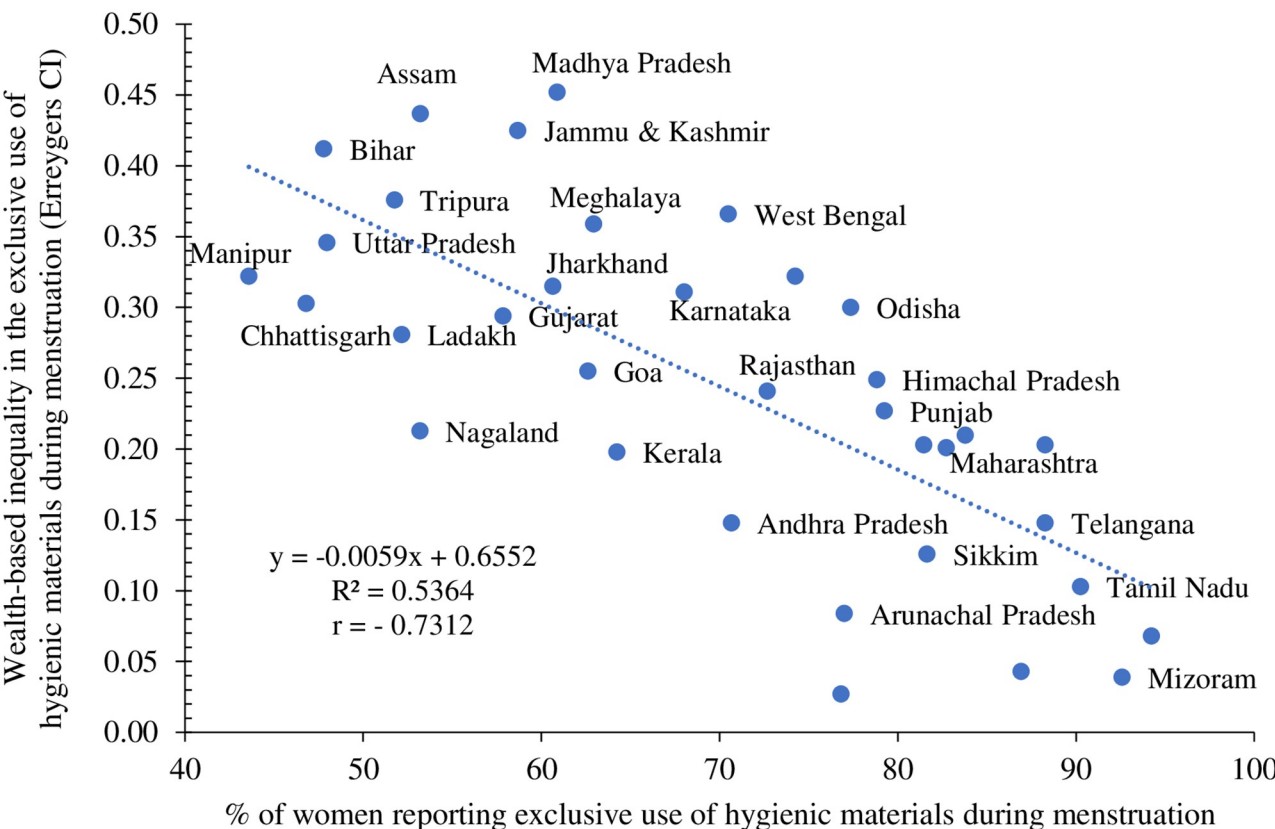

**Fig 6. Association between level of exclusive use of hygienic materials and wealth-based inequality in the same among urban women aged 15–24 years across the states & UTs\* of India, NFHS-5, 2019–21.** Note: \* Excluding Lakshadweep which was an outlier.

**Table 3. Decomposition of wealth-based inequality in the exclusive use of hygienic materials among urban women aged 15–24 years, NFHS-5, 2019–21.**

| Variables | Elasticity | Concentration Index | Contribution | Contribution (in %) |
|---|---|---|---|---|
| Age at menarche | 0.029 | 0.017 | 0.000 | 0.23 |
| Education | 0.721 | 0.139 | 0.100 | 47.66 |
| Religion | 0.217 | 0.019 | 0.004 | 1.98 |
| SC/ST | -0.035 | -0.186 | 0.006 | 3.08 |
| Mass media exposure | 0.465 | 0.145 | 0.068 | 32.05 |
| Child marriage | -0.025 | -0.118 | 0.003 | 1.40 |
| Type of home | -0.038 | -0.106 | 0.004 | 1.92 |
| North | 0.113 | 0.171 | 0.019 | 9.22 |
| Central | -0.060 | -0.014 | 0.001 | 0.40 |
| East | 0.034 | -0.236 | -0.008 | -3.76 |
| West | 0.099 | 0.087 | 0.009 | 4.09 |
| South | 0.147 | 0.025 | 0.004 | 1.75 |
| North-east | 0.000 | -0.033 | 0.000 | 0.00 |
| **Explained CI** | | | **0.211** | **100.00** |
| **Actual CI** | | | **0.302** | |
| **Residual** | | | **0.091** | |

material use during menstruation. Women with higher education are generally more aware of modern hygienic materials and their benefits. They are more knowledgeable of the risks of using unhygienic materials during menstruation. Additionally, they frequently have greater financial independence and decision-making power within their households (24,25,27). Our study also revealed a pro-rich inequality in educational attainment, which significantly contributed to the inequality in the exclusive use of hygienic materials among urban women.

Young women in urban areas exposed to at least one type of mass media were more likely to use hygienic materials than those without exposure. It is argued that exposure to mass media increases awareness and dissemination of knowledge about existing menstrual hygiene programs and policies. As a reliable source of information, the mass media raises awareness about the health benefits of using hygienic materials and broaden knowledge of hygienic materials available on the market and those subsidized by the government [26, 28, 45]. It can also alter women's attitudes towards sanitary napkins, which are considered hygienic [46, 47]. Our research discovered a pro-rich concentration of mass media exposure among urban women, which played a significant role in the existing inequality in the outcome.

Regional disparities turned out to be a contributing factor to the inequality in the exclusive use of hygienic materials among young urban women. Inequality in the outcome has been exacerbated by socioeconomic disparity that has spread across the country. The prevalence of exclusive use of hygienic materials among urban women is higher in the southern and northern regions than among urban women in the central region. These findings are similar to previous studies, which have noted substantial differences in the use of hygienic materials between the southern and central Indian states [19, 25, 26, 48, 49]. The southern states of India are generally more socioeconomically advanced, with better access to public healthcare and better-managed subsidized sanitary pad programs. The exclusive use is deficient in the central region because these states are characterized by low socioeconomic development, high levels of urban poverty, and an inefficient public healthcare system. These reasons, coupled with the strong presence of social taboos, could be behind the lower exclusive use of hygienic materials among poor urban women in these regions (24,25).

In the eighties and the early nineties of the last century, menstrual health was not on the government agenda in India [15]. When the Reproductive and Child Health Programme (RCH) was launched in the late nineties, menstrual health got attention but was not thoroughly addressed [50]. Menstrual health gained more attention from the government in the first decade of the current century when the Accredited Social Health Activists (ASHA), the village-level health workers recruited under the National Rural Health Mission (2005), were given the responsibility of menstrual health management [15]. Both the Central and the State Governments of India have been making concerted efforts to promote menstrual hygiene among women ever since. The Ministry of Health and Family Welfare of the Central Government launched the Menstrual Hygiene Scheme (MHS) in 2011 under *Rashtriya Kishor Swasthya Karyakram* (RKSK) [51]. It was India's first comprehensive adolescent health program to raise menstrual hygiene awareness by promoting access to and use of high-quality sanitary napkins and ensuring safe, environmentally sustainable disposal of sanitary napkins [51, 52]. However, a study revealed that MHS is not operating per the standards in some Indian states, such as Haryana, Madhya Pradesh, Maharashtra, and Uttarakhand, mainly due to procurement challenges [52]. In 2018, the Central Government launched a 100% oxo-biodegradable sanitary napkins brand, '*Suvidha*', at a subsidized price at *Jan Aushadhi Kendras* (government-subsidized pharmacies) in various cities across the country. However, a report revealed that they often ran out of supply of these subsidized sanitary napkins [53].

Tamil Nadu has considerably low socioeconomic inequality in the exclusive use of hygienic materials among urban women, possibly because the state government has been providing 20

sanitary napkins free of cost every month to women since 2011 under the scheme *Pudhu Yugam* (New Era) [54]. In some districts of Tamil Nadu, Andhra Pradesh, and Telangana, schools, in partnership with local NGOs, have installed sanitary napkin vending machines that dispense locally-produced napkins at a subsided rate [15, 55]. These efforts have helped reduce wealth-based inequality in the use of hygienic materials, as poor girls/women can easily access and utilize sanitary products.

Both the Central and the State governments have recently initiated several programs to provide subsidized sanitary napkins to girls and women. However, most of them have been implemented on a pilot basis. These programs need to be scaled up further to expand their reach to the poorest of the poor households in urban areas. In addition to providing sanitary napkins at a subsidized cost, governments should also make efforts to spread awareness about the benefits of using hygienic materials. Among other measures that could promote the use of hygienic materials among girls and women include making menstrual hygiene materials tax free and improving the quality of the absorbent materials [56, 57].

This study has some limitations. Firstly, we could not determine how accessibility and affordability of such products are related to economic inequality as the NFHS-5 does not contain any questions regarding the affordability and availability of hygienic materials. Secondly, as the data used in this study is cross-sectional, we could not find any causal factors of this inequality. We needed experimental data to identify causality. Thirdly, due to data limitations, this research was unable to account for the contribution of supply-side factors affecting the exclusive use of hygienic materials during menstruation. Finally, this research did not investigate the past trends and patterns of wealth-based inequalities in the exclusive use of hygienic materials in urban India. Further research should be done to investigate this issue in detail. Future research might explore the issue of affordability and accessibility of hygienic materials among urban women in India.

## Conclusion

The exclusive use of hygienic materials during menstruation is still considerably low among urban women in many states of India. This study has demonstrated a pro-rich inequality in the exclusive use of hygienic materials that needs to be addressed with appropriate programmatic interventions. The wealth-based inequality in the exclusive use of hygienic materials varies considerably across Indian states. Addressing this spatial variation in wealth-based inequality in the exclusive use of hygienic materials would require state-specific interventions. The state-wise estimates of wealth-based inequality provided in the study could be useful in tailoring state-specific policies and programs to promote the use of hygienic materials among underprivileged urban women. In addition, this study has also identified women's education and mass media exposure as the major contributors to existing wealth-based inequality in the exclusive use of hygienic materials. Therefore, future programs and policies should focus on poor and uneducated women in urban areas and strengthen existing efforts to create awareness around menstruation through different mass media.

## Supporting information

**S1 Table. Erreygers normalised CI for Indian states and UTs in the exclusive use of hygienic materials among urban women aged 15–24 years in India, NFHS-5, 2019–21.** (DOCX)

## Acknowledgments

We would like to acknowledge and give warmest thanks to Professor Guido Erreygers, Department of Economics, University of Antwerp (Belgium) and Dr. Avijit Debnath, Assistant Professor of Economics, Assam University (India) who made this work possible. We benefited much from their guidance and direction while we drafted the methodology section of this article.

## Author Contributions

**Conceptualization:** Aditya Singh.

**Data curation:** Mahashweta Chakrabarty.

**Formal analysis:** Aditya Singh, Mahashweta Chakrabarty, Shivani Singh, Sourav Chowdhury.

**Investigation:** Shivani Singh, Diwakar Mohan, Rakesh Chandra.

**Methodology:** Aditya Singh, Mahashweta Chakrabarty, Shivani Singh, Diwakar Mohan.

**Project administration:** Aditya Singh, Shivani Singh.

**Supervision:** Aditya Singh, Mahashweta Chakrabarty, Shivani Singh, Diwakar Mohan, Rakesh Chandra.

**Validation:** Aditya Singh, Mahashweta Chakrabarty.

**Visualization:** Aditya Singh, Mahashweta Chakrabarty, Sourav Chowdhury.

**Writing – original draft:** Aditya Singh, Mahashweta Chakrabarty, Rakesh Chandra, Sourav Chowdhury.

**Writing – review & editing:** Aditya Singh, Mahashweta Chakrabarty, Shivani Singh, Diwakar Mohan, Rakesh Chandra, Sourav Chowdhury.

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
