## [Decision Letter · Decision Letter 0]

25 Jul 2022

PONE-D-22-18160Wealth-Based Inequality in the Exclusive Use of Hygienic Absorbents during Menstruation in Urban India: A Decomposition and Spatial ApproachPLOS ONE

Dear Dr. Singh,

Thank you for submitting your manuscript to PLOS ONE. After careful consideration, we feel that it has merit but does not fully meet PLOS ONE’s publication criteria as it currently stands. Therefore, we invite you to submit a revised version of the manuscript that addresses the points raised during the review process.

We look forward to receiving your revised manuscript.

Kind regards,

Vijayaprasad Gopichandran

Academic Editor

PLOS ONE

Journal Requirements:

2.Please provide additional details regarding participant consent. In the ethics statement in the Methods and online submission information, please ensure that you have specified what type you obtained (for instance, written or verbal, and if verbal, how it was documented and witnessed). If your study included minors, state whether you obtained consent from parents or guardians. If the need for consent was waived by the ethics committee, please include this information.

3. We note that Figure 2 in your submission contain [map/satellite] images which may be copyrighted. All PLOS content is published under the Creative Commons Attribution License (CC BY 4.0), which means that the manuscript, images, and Supporting Information files will be freely available online, and any third party is permitted to access, download, copy, distribute, and use these materials in any way, even commercially, with proper attribution. For these reasons, we cannot publish previously copyrighted maps or satellite images created using proprietary data, such as Google software (Google Maps, Street View, and Earth). For more information, see our copyright guidelines: http://journals.plos.org/plosone/s/licenses-and-copyright.

Reviewers' comments:

Reviewer's Responses to Questions

**Comments to the Author**

1. Is the manuscript technically sound, and do the data support the conclusions?

Reviewer #1: No

2. Has the statistical analysis been performed appropriately and rigorously? 

Reviewer #1: I Don't Know

3. Have the authors made all data underlying the findings in their manuscript fully available?

Reviewer #1: Yes

4. Is the manuscript presented in an intelligible fashion and written in standard English?

Reviewer #1: No

5. Review Comments to the Author

Reviewer #1: This study provides information on a neglected topic - menstrual health and hygiene which has been under researched. I applaud the authors for pursuing this topic and highlight a few areas of concern below:

(1) Abstract background and throughout - kindly use hygienic materials or hygienic products as one of the products mentions - menstrual cups - are not an absorbent. Kindly rectify throughout the manuscript.

(2) Abstract - methods: rather than state "using a relatively new, modified technique" kindly summarize the technique here.

(3) Abstract - methods: kindly note your study population and location

(4) Abstract - methods: it is unclear what is meant by "decomposed the CI to address the key determinants of inequality" - please explain

(5) Abstract - conclusion: Is it assumed that women from poorer households can afford hygienic menstrual products and that if they had access to media and education they would be able to purchase these?

(6) Introduction - just to repeat that menstrual cups are not an absorbent so

(7) Introduction - please check for grammar: "Several studies have reported that rapid urbanization not only boosts economic growth [comma] but also results [IN]....With increasing urbanization... where THE majority of households are poor. The living CONDITIONS .... ARE .... the rest of the WEALTHIER .... [please check throughout the subsequent paragraphs as well]

(8) Please provide references to support the sentence opening "Indian society, by virtue...."

(9) Sentence opening - Reproductive and gynaecological diseases are the fifth leading cause of reduced

disability-adjusted life year (DALY).... should this say "in India" ? This sentence follows the argument that women in households must prioritize the household but the age range listed here is for adolescent girls and young women. Should we assume that the prior sentences should say women and girls?

(10) It seems a little overstated to say that menstrual health is the "most important" aspect of girls health to reduce the burden of reproductive and gynaecological disease. Others would argue that maternal health (during pregnancy) or unsafe sex would be "the most" important aspects that would improve reproductive health in girls. I would try to better qualify this sentence and let menstrual health stand alone -- something like "Menstrual health and hygiene is a critical need for girls and women; menarche is the biological event that marks when a girls enters her reproductive years and after which she can become pregnant. However, menstrual health has remained largely ignored in reproductive health research and warrants special attention in its contribution to girls' and women's burden of reproductive disease." for example (please feel free to use your own language).

(11) Next paragraph to note again that menstrual cups are not an absorbent - please use "products" or "materials"

(12) Please provide a reference for sentence beginning: "Ensuring universal access to menstrual hygienic absorbents ..."

(13) Sentence unclear -- "However, little is known about the extent.... in the outcome." What is meant by "in the outcome" do you mean to say "However, little is known on the extent and impact of wealth-based inequality on use of hygienic products among urban women in India."

(14) Last sentence in introduction -- This study helps to measure ... is very unclear. Also the final clause "which should be an integral part" is hanging and does not belong in your pre-methods statement.

(15) To note again please have someone check your grammar -- Methods first paragraph example: "We used data from the latest [OMIT FIFTH] round of the National Family Health Survey (NFHS-5) WHICH WAS conducted BETWEEN 2019 AND 2021. THE NFHS is a large[OMIT SCALE], nationally representative survey that provides

data on India’s population, health, and nutrition STATUS. [OMIT and its states] (27)... interviewed 724,115 women of reproductive age [OMIT group] (15-49 years) from 636,699 households. WE EXCLUDED 544,580 rural women [OMIT excluded] DUE TO OUR FOCUS ON urban women IN India. [OMIT Furhter] From THESE 179,535 [omit remaining] women, 124,974 urban women aged 25-49 years WERE excluded because OF MISSING menstrual hygiene DATAPOINTS. THE remaining, 54,561 urban women aged 15-24 from 28 states, 8 union territories and 707 districts were included in

the analysis.

(16) In your dependent variable no mention of menstrual cups is made - however throughout you mention this as a hygienic option. You may wish to exclude cups from this manuscript and maintain absorbents as your word of choice given they were not captured in the NFHS.

(17) In your dependent variable you code as 2 "both disposable and reusable absorbents," unclear why these would be considered unhygienic?

(18) Methods - why were only some women asked the menstruation related questions?

(19) Table 1 -- your definition for age at menarche is confusing - do you remove all women who had menarche prior to the age of 13? Please also check your grammar on this table

(20) Again definition of other measures - education do you mean they got a 1 if completed secondary or above and a 0 if did not reach secondary completion or below? Please clarify your definitions

(21) Table 1 - SC/ST is this a categorical variable?

(22) Table 1 - type of home is "marital home" vs "other" ?

(23) Table 1 - your main exposure variable household wealth is missing in this table - please add including the dimensions for the PCA

(24) Unclear in your results you say central region had lowest exclusive use of absorbents followed by stating Manipur has been facing the worst situation. Is this a sub-region within the region? For those not familiar with India's geography this is not easy to follow and needs a better introduction.

(25) Discussion - I'm still unclear on if the study took into consideration how wealth-based inequality was a direct barrier to purchase because women did not have money to buy hygienic menstrual wear rather than their education or exposure to media.

(26) Discussion - you state "Only providing sanitary napkins at subsidized cost won’t help in reducing economic inequality

in hygienic absorbents use unless the women are provided with the knowledge of benefits of using hygienic absorbents; availability, safety of hygienic materials, referral and access to health services, sanitation and washing facilities, positive social norms, safe and hygienic disposal, advocacy and policy." Where in your results is this statement supported? How do we know that removing barriers to access such as giving women pads would not improve their ability to use them? Have you looked at whether women who were the most impoverished could afford hygienic products?

(27) In your conclusion you state -- There is a tendency to consider that urban areas have better healthcare facilities , improved health outcome, and low wealth-based inequality as compared to its rural counter parts. However, this fact has been proved wrong by several research in India, as with rapid urbanization India, urban poverty is increasing and urban areas are suffering from sanitation, water, drainage, and disposal problems." -- However, no where in this manuscript to you compare urban to rural women. How do you support this statement?

6. PLOS authors have the option to publish the peer review history of their article (what does this mean?). If published, this will include your full peer review and any attached files.

Reviewer #1: No

---

## [Author Response · Author response to Decision Letter 0]

18 Oct 2022

Dear Dr. Vijayaprasad Gopichandran,

We would like to thank you for providing us an opportunity to revise our paper entitled “Wealth-Based Inequality in the Exclusive Use of Hygienic Absorbents during Menstruation in Urban India: A Decomposition and Spatial Approach” (PONE-D-22-18160). We extend our heartfelt thanks to the editor and the reviewer for their valuable time and helpful comments and suggestions. The manuscript has been revised in the light of the concerns raised in the review report. We have addressed all the issues raised and incorporated all the comments and suggestions made by the reviewers. Our point-to-point response to the comments, on which the reviewers needed further clarification from us or suggested modifications to be made in the manuscript, is given below.

 Reviewer(s)’ comments to the author

Reviewer: 1

This study provides information on a neglected topic - menstrual health and hygiene which has been under researched. I applaud the authors for pursuing this topic and highlight a few areas of concern below.

Thank you, sir, for your encouraging words. We are indebted to you for investing your valuable time and making highly useful suggestions to make this paper publishable.

Comment#1: Abstract background and throughout - kindly use hygienic materials or hygienic products as one of the products mentions - menstrual cups - are not an absorbent. Kindly rectify throughout the manuscript.

Response#1: Thank you, sir, for pointing out this issue. As per your suggestion, we have, in the abstract background and throughout the manuscript, replaced the term ‘hygienic absorbents’ with ‘hygienic materials.’

Comment#2: Abstract - methods: rather than state "using a relatively new, modified technique" kindly summarize the technique here.

Response#2: In the ‘Abstract – methods’ in the place of “using a relatively new, modified technique” we have mentioned the name of the technique used for decomposition analysis. (Line 45-47)

Comment#3: Abstract - methods: kindly note your study population and location

Response#3: Thank you for pointing out this shortcoming. In the revised manuscript, we have included the study population and location of the study as per your suggestion . 

Comment#4: Abstract - methods: it is unclear what is meant by "decomposed the CI to address the key determinants of inequality" - please explain.

Response#4: According to Wagstaff, van Doorslaer, and Watanabe (2003), the health concentration index can be decomposed into the contributions of individual factors to income-related health inequality, in which each contribution is the product of the sensitivity of health variable with respect to that factor and the degree of income-related inequality in that factor. 

In our study, we have computed the concentration index to measure the wealth-based inequality in the exclusive use of hygienic materials and then decomposed this wealth-based inequality into the contributions of individual factors such as age at menarche, child marriage, respondent’s level of education, religion, social groups, type of home, mass media exposure to the wealth-based inequality.

For decomposition, we used Erreygers decomposition method which is a regression-based decomposition, in which we decompose existing wealth-based inequality (in the use of hygienic materials across different wealth-groups) to understand how wealth-based inequality in educational attainment, mass media exposure, religion, social groups, and other factors contribute to wealth-based inequality in the use of hygienic materials during menstruation.

To make the addressed sentence clearer and understandable, we have rewritten it as follows: Further, we decomposed the Erreygers CI to estimate the relative contribution of covariates to wealth-based inequality in the use of hygienic materials during menstruation.

We hope this clarifies what we meant by decomposition of inequality. 

Comment#5: Abstract - conclusion: Is it assumed that women from poorer households can afford hygienic menstrual products and that if they had access to media and education, they would be able to purchase these?

Response#5: Thank you for pointing out this confusing text. What we wanted to state was that the existing economic inequalities in access to mass media and educational level among urban women may be exacerbating the economic inequality in the exclusive use of hygienic materials. Hence, existing and future programs and interventions should focus on women with little or no education and low mass media exposure. 

Comment#6: Introduction - just to repeat that menstrual cups are not an absorbent so

Response#6: Yes, thank you for pointing this out. In the revised manuscript, following your suggestion, we have used the term ‘hygienic materials’ instead of ‘absorbents.’

Comment#7: Introduction - please check for grammar: "Several studies have reported that rapid urbanization not only boosts economic growth [comma] but also results [IN].... With increasing urbanization... where THE majority of households are poor. The living CONDITIONS .... ARE .... the rest of the WEALTHIER .... [please check throughout the subsequent paragraphs as well]

Response#7: We have corrected the grammar of mentioned sentences, also rewritten some of the sentences to make the text more comprehensible. 

Comment#8: Please provide references to support the sentence opening "Indian society, by virtue...."

Response#8: We have added suitable reference to support the above-mentioned sentence.

Comment#9: Sentence opening - Reproductive and gynaecological diseases are the fifth leading cause of reduced disability-adjusted life year (DALY).... should this say "in India”? This sentence follows the argument that women in households must prioritize the household but the age range listed here is for adolescent girls and young women. Should we assume that the prior sentences should say women and girls? 

Response#9: According to the Global Burden of Diseases study cited in support of the concerned sentence in the original manuscript, reproductive and gynecological diseases were the fifth leading cause of reduced disability-adjusted life year (DALY) in the age group 10-24 years for both males and females in 2019 across the 204 countries (including India). 

In the revised manuscript, we have rewritten the sentence and have referred to the GBD data for women of reproductive age (15-49) in India, rather than the world. According to the GBD data, gynecological diseases are the leading cause of disease burden among women of reproductive age in India, contributing about 6% of the total disability-adjusted life years (DALYs). Reference: Institute for Health Metrics and Evaluation. Global Burden of Disease (GBD) [Internet]. Institute for Health Metrics and Evaluation. 2019. Available from: https://vizhub.healthdata.org/gbd-compare/

We hope that the revised text is not confusing anymore. 

Comment#10: It seems a little overstated to say that menstrual health is the "most important" aspect of girls’ health to reduce the burden of reproductive and gynaecological disease. Others would argue that maternal health (during pregnancy) or unsafe sex would be "the most" important aspects that would improve reproductive health in girls. I would try to better qualify this sentence and let menstrual health standalone -- something like "Menstrual health and hygiene is a critical need for girls and women; menarche is the biological event that marks when a girl enters her reproductive years and after which she can become pregnant. However, menstrual health has remained largely ignored in reproductive health research and warrants special attention in its contribution to girls' and women's burden of reproductive disease." for example (please feel free to use your own language).

Response#10: Thank you for pointing out the issue. We have revised the lines as per your suggestion.

Comment#11: Next paragraph to note again that menstrual cups are not an absorbent - please use "products" or "materials"

Response#11: As per NFHS-5 report we have replaced the term ‘absorbents’ with ‘materials’ throughout the revised text.

Comment#12: Please provide a reference for sentence beginning: "Ensuring universal access to menstrual hygienic absorbents ..."

Response#12: We have provided a suitable citation in support of our statement. 

Comment#13: Sentence unclear -- "However, little is known about the extent.... in the outcome." What is meant by "in the outcome" do you mean to say "However, little is known on the extent and impact of wealth-based inequality on use of hygienic products among urban women in India."

Response#13: Yes. With the term ‘outcome’, we meant ‘wealth-based inequality in use of hygienic materials’. Since the word ‘outcome’ was creating confusion, we have revised the text as per your suggestion.

Comment#14: Last sentence in introduction -- This study helps to measure ... is very unclear. Also, the final clause "which should be an integral part" is hanging and does not belong in your pre-methods statement.

Response#14: Thanks for flagging this issue. We have revised the text and now clearly mentioned that the main aim of this study is to measure the wealth-based inequality in the exclusive use of hygienic materials among urban women and decompose this inequality in its contributing factors. We have removed the clause "which should be an integral part" in the revised version of the manuscript. 

Comment#15: To note again please have someone check your grammar -- Methods first paragraph example: "We used data from the latest [OMIT FIFTH] round of the National Family Health Survey (NFHS-5) WHICH WAS conducted BETWEEN 2019 AND 2021. THE NFHS is a large [OMIT SCALE], nationally representative survey that provides data on India’s population, health, and nutrition STATUS. [OMIT and its states] (27) ... interviewed 724,115 women of reproductive age [OMIT group] (15-49 years) from 636,699 households. WE EXCLUDED 544,580 rural women [OMIT excluded] DUE TO OUR FOCUS ON urban women IN India. [OMIT Further] From THESE 179,535 [omit remaining] women, 124,974 urban women aged 25-49 years WERE excluded because OF MISSING menstrual hygiene DATAPOINTS. THE remaining, 54,561 urban women aged 15-24 from 28 states, 8 union territories and 707 districts were included in the analysis.

Response#15: Thanks you sir/madam for pointing out these grammatical and spelling mistakes. Following your suggestions, we have revised the entire text mentioned by you. 

Comment#16: In your dependent variable no mention of menstrual cups is made - however throughout you mention this as a hygienic option. You may wish to exclude cups from this manuscript and maintain absorbents as your word of choice given, they were not captured in the NFHS.

Response#16: Thank you sir for pointing out this very important issue. We did include menstrual cup in our analysis but somehow forgot to mention menstrual cup in our dependent variables section. We have mentioned this in ‘outcome variable’ section of the revised manuscript. 

Comment#17: In your dependent variable you code as 2 "both disposable and reusable absorbents," unclear why these would be considered unhygienic?

Response#17: For more clarity we have removed the phrase "both disposable and reusable absorbents," and we have rewritten this section. I hope now the modified text makes more sense. 

According to NFHS 5 report, sanitary napkins, locally prepared napkins, tampons, and menstrual cups are considered as hygienic materials. Materials other than these (i.e., clothes, nothing, others), used during menstruation are considered as unhygienic.

As our outcome variable of interest is “exclusive use of hygienic materials”; this is a binary variable. A woman is considered “an exclusive user of hygienic materials” if she uses hygienic materials only (coded as ‘1’). Any woman who either uses unhygienic materials or a combination of hygienic and unhygienic materials is considered “not an exclusive user of hygienic materials” (coded as ‘0’). 

Women who use a combination of hygienic and unhygienic materials are considered unhygienic in our study, because previous literature have reported women who used hygienic and unhygienic materials simultaneously were susceptible to RTI as compared to those who solely used hygienic materials during menstruation. 

This variable has been defined in this way in many previous studies in India.

Ref: Anand, E., J. Singh, and S. Unisa. Menstrual hygiene practices and its association with reproductive tract infections and abnormal vaginal discharge among women in India. Sex Reprod Healthc. 2015; 6 (4): 249–54. Available from: https://www.researchgate.net/publication/279312494_Menstrual_hygiene_practices_and_its_association_with_reproductive_tract_infections_and_abnormal_vaginal_discharge_among_women_in_India

Ram U, Pradhan MR, Patel S, Ram F. Factors associated with disposable menstrual absorbent use among young women in India. Int Perspect Sex Reprod Health. 2020;46(October):223–34. Available from: https://www.guttmacher.org/journals/ipsrh/2020/10/factors-associated-disposable-menstrual-absorbent-use-among-young-women-india

Bs, D. A. McKenna and N. Swarnalatha. “Prevalence of RTI/STI among reproductive age women (15-49) years in urban slums of Tirupati Town Andhra Pradesh.” Health and population; perspectives and issues 30 (2007). Available from: https://www.semanticscholar.org/paper/Prevalence-of-RTI%2FSTI-among-reproductive-age-women-Bs-Swarnalatha/18e87dfc3f1b0b9fce6c5e07fa93985e97952cf4

Comment#18: Methods - why were only some women asked the menstruation related questions?

Response#18: In NFHS-5, only women aged 15-24 years are asked the menstruation related questions. Neither the NFHS report nor the IIPS website explains why only women aged 15–24 was asked menstrual hygiene questions.

Comment#19: Table 1 -- your definition for age at menarche is confusing - do you remove all women who had menarche prior to the age of 13? Please also check your grammar on this table

Response#19: I have corrected the grammar of the sentence and clearly mentioned that ‘age at menarche’ indicates the age of onset of first menstrual period of a women. 

No, we haven’t removed any woman who had menarche prior to the age of 13. Urban women who received their first period at 13 years and above is coded as “1” and if the women received her first period before 13 years, then coded as “0”. 

Comment#20: Again, definition of other measures - education do you mean they got a 1 if completed secondary or above and a 0 if did not reach secondary completion or below? Please clarify your definitions.

Response#20: Education is categorized as a binary variable. 15-24 years urban women who have completed their education till secondary level and above are coded as “1” and as “0” if they did not reach secondary completion or below.

Comment#21: Table 1 - SC/ST is this a categorical variable?

Response#21: SC/ST is a categorical variable having two categories. Women belonging to Scheduled Caste or Scheduled Tribe are coded as “1” and those belonging to Other Backward Classes or Others are coded as ‘0’.

Comment#22: Table 1 - type of home is "marital home" vs "other”?

Response#22: On the basis of the woman's relationship with the household head, the type of home in which she resided was categorized as ‘marital home’ (wife, daughter-in-law, or sister-in-law of the household head) coded as “1” and coded as “0” otherwise (daughter, granddaughter, or niece of the household head, nonrelatives such as domestic servants working in the household, orphans, deserted young women).

Comment#23: Table 1 - your main exposure variable household wealth is missing in this table - please add including the dimensions for the PCA

Response#23: Thank you for pointing out this issue. The wealth score variable that we have used in this study is already provided in the dataset. Table 1 of the revised manuscript now provides a description of this variable based on the NFHS-5 national report. Since this PCA was not conducted by us, we are unable to provide the details of the PCA. We have mentioned whatever information we could get from the NFHS-5 national report.

Comment#24: Unclear in your results you say central region had lowest exclusive use of absorbents followed by stating Manipur has been facing the worst situation. Is this a sub-region within the region? For those not familiar with India's geography this is not easy to follow and needs a better introduction.

Response#24: At first, we have divided India into 6 regions to observe geographical disparities in the prevalence of using hygienic methods. 

The northern region includes Jammu & Kashmir, Ladakh, Himachal Pradesh, Punjab, Rajasthan, Haryana, Uttarakhand, Chandigarh (Union Territory - UT) and Delhi

The central region includes the states of Uttar Pradesh, Madhya Pradesh and Chhattisgarh.

The eastern region includes the states of Bihar, Jharkhand, West Bengal and Odisha. 

The western includes the states of Gujarat, Maharashtra, Goa and UTs of Dadra & Nagar Haveli and Daman & Diu.

The southern region includes the states of Kerala, Karnataka, Andhra Pradesh, Tamil Nadu and the UTs of Andaman & Nicobar Islands, Pondicherry and Lakshadweep

The northeastern region includes the states of Sikkim, Assam, Meghalaya, Manipur, Mizoram, Nagaland, Tripura, and Arunachal Pradesh. 

According to the findings of our study, the exclusive use of hygienic materials among urban women is lowest in the central region (50.9%) of India.

However, among the 28 states of India (state level analysis), it is evident that Manipur is facing the worst situation (43.6%). 

To help the reader understand the division of the regions of India, we have given a full list of states included in each region in the variable list given in Table 1.

We have also revised the text in this entire section to clear up any confusion (see lines 224-231)

Comment#25: Discussion - I'm still unclear on if the study took into consideration how wealth-based inequality was a direct barrier to purchase because women did not have money to buy hygienic menstrual wear rather than their education or exposure to media.

Response#25: Our study does not consider how wealth inequality was a direct barrier to purchase because women did not have enough money to buy hygienic menstrual products; rather, our study aims to measure how the exclusive use of hygienic materials is concentrated within a specific wealth group or unevenly distributed across different household wealth categories. 

Our analysis revealed that the exclusive use of hygienic materials is concentrated among the wealthier households. This could be because a number of predictors of the exclusive use of hygienic materials, including education and mass media exposure, are concentrated among the wealthier households. 

As NFHS does not provide any data on the purchasing power, or affordability of hygienic materials, we could not analyze the problems related to affordability of the hygienic martials, or how low wealth or income was a direct barrier to purchase.

Comment#26: Discussion - you state "Only providing sanitary napkins at subsidized cost won’t help in reducing economic inequality in hygienic absorbents use unless the women are provided with the knowledge of benefits of using hygienic absorbents; availability, safety of hygienic materials, referral and access to health services, sanitation and washing facilities, positive social norms, safe and hygienic disposal, advocacy and policy." Where in your results is this statement supported? How do we know that removing barriers to access such as giving women pads would not improve their ability to use them? Have you looked at whether women who were the most impoverished could afford hygienic products?

Response#26: Thank you for pointing out this very important issue in the conclusion section. Yes, the text was unnecessary and not supported by the findings, hence keeping in mind your suggestion, we have revised this section thoroughly to reflect the findings. We hope the revised text reads much better now and conveys the intended meaning. 

Comment#27: In your conclusion you state -- There is a tendency to consider that urban areas have better healthcare facilities, improved health outcome, and low wealth-based inequality as compared to its rural counter parts. However, this fact has been proved wrong by several research in India, as with rapid urbanization India, urban poverty is increasing and urban areas are suffering from sanitation, water, drainage, and disposal problems." -- However, nowhere in this manuscript to you compare urban to rural women. How do you support this statement?

Response#27: Thank you once again for highlighting this issue. We have revised the text thoroughly in the conclusion section keeping in view the important queries raised by your comment #26 and #27. The revised text in the conclusion is in line of our findings. Any sentence not supported by our findings has been removed. We hope this addresses your concern. 

Once again, we appreciate your insightful suggestions and comments. The reviewer's helpful suggestions have allowed us to make significant revisions to the paper, which we feel greatly improved the article. In addition, we have slightly altered the title of our revised manuscript to better reflect its contents (New/modified title: Wealth-Based Inequality in the Exclusive Use of Hygienic Materials During Menstruation Among Young Women in Urban India.)

After making all of the revisions recommended by the reviewer, we are hopeful that this manuscript is suitable to be considered to your prestigious journal. 

Sincerely yours,

Aditya Singh 

Mahashweta Chakrabarty

Shivani Singh

Diwakar Mohan

Rakesh Chandra

Sourav Chowdhury

---

## [Editor Report · Decision Letter 1]

20 Oct 2022

Wealth-Based Inequality in the Exclusive Use of Hygienic Materials During Menstruation Among Young Women in Urban India

PONE-D-22-18160R1

Dear Dr. Singh,

We’re pleased to inform you that your manuscript has been judged scientifically suitable for publication and will be formally accepted for publication once it meets all outstanding technical requirements.

Kind regards,

Vijayaprasad Gopichandran

Academic Editor

PLOS ONE
---

## [Editor Report · Acceptance letter]

17 Nov 2022

PONE-D-22-18160R1 

Wealth-Based Inequality in the Exclusive Use of Hygienic Materials During Menstruation Among Young Women in Urban India 

Dear Dr. Singh:

I'm pleased to inform you that your manuscript has been deemed suitable for publication in PLOS ONE. Congratulations! Your manuscript is now with our production department. 

Kind regards, 

on behalf of

Dr. Vijayaprasad Gopichandran 

Academic Editor

PLOS ONE